# Immunogenicity and In Vivo Protective Effects of Recombinant Nucleocapsid-Based SARS-CoV-2 Vaccine Convacell^®^

**DOI:** 10.3390/vaccines11040874

**Published:** 2023-04-20

**Authors:** Sevastyan O. Rabdano, Ellina A. Ruzanova, Iuliia V. Pletyukhina, Nikita S. Saveliev, Kirill L. Kryshen, Anastasiia E. Katelnikova, Petr P. Beltyukov, Liliya N. Fakhretdinova, Ariana S. Safi, German O. Rudakov, Sergei A. Arakelov, Igor V. Andreev, Ilya A. Kofiadi, Musa R. Khaitov, Rudolf Valenta, Daria S. Kryuchko, Igor A. Berzin, Natalia S. Belozerova, Anatoly E. Evtushenko, Viktor P. Truhin, Veronika I. Skvortsova

**Affiliations:** 1Saint Petersburg Scientific Research Institute of Vaccines and Serums of the Federal Medical-Biological Agency of Russia (SPbSRIVS), St. Petersburg 198320, Russia; 2RMC “Home of Pharmacy“ JSC, Kuzmolovsky 188663, Russia; 3Scientific Research Institute of Hygiene, Occupational Pathology and Human Ecology of the Federal Medical-Biological Agency of Russia (SRIHOPHE), Kuzmolovsky 188663, Russia; 4National Research Center Institute of Immunology (NRCII), Federal Medical-Biological Agency of Russia, Moscow 115522, Russia; 5Department of Immunology, N.I. Pirogov Russian National Research Medical University, Ministry of Health of the Russian Federation, Moscow 117997, Russia; 6Department of Pathophysiology and Allergy Research, Center for Pathophysiology, Infectiology and Immunology, Medical University of Vienna, 1090 Vienna, Austria; 7Laboratory of Immunopathology, Department of Clinical Immunology and Allergology, I.M. Sechenov First Moscow State Medical University, Moscow 119435, Russia; 8Karl Landsteiner University of Health Sciences, 3500 Krems, Austria; 9Federal Medical-Biological Agency of Russia, Moscow 125310, Russia

**Keywords:** protein N, nucleocapsid, vaccine, SARS-CoV-2, COVID-19

## Abstract

The vast majority of SARS-CoV-2 vaccines which are licensed or under development focus on the spike (S) protein and its receptor binding domain (RBD). However, the S protein shows considerable sequence variations among variants of concern. The aim of this study was to develop and characterize a SARS-CoV-2 vaccine targeting the highly conserved nucleocapsid (N) protein. Recombinant N protein was expressed in *Escherichia coli*, purified to homogeneity by chromatography and characterized by SDS-PAGE, immunoblotting, mass spectrometry, dynamic light scattering and differential scanning calorimetry. The vaccine, formulated as a squalane-based emulsion, was used to immunize Balb/c mice and NOD SCID gamma (NSG) mice engrafted with human PBMCs, rabbits and marmoset monkeys. Safety and immunogenicity of the vaccine was assessed via ELISA, cytokine titer assays and CFSE dilution assays. The protective effect of the vaccine was studied in SARS-CoV-2-infected Syrian hamsters. Immunization induced sustainable N-specific IgG responses and an N-specific mixed Th1/Th2 cytokine response. In marmoset monkeys, an N-specific CD4^+^/CD8^+^ T cell response was observed. Vaccinated Syrian hamsters showed reduced lung histopathology, lower virus proliferation, lower lung weight relative to the body, and faster body weight recovery. Convacell^®^ thus is shown to be effective and may augment the existing armamentarium of vaccines against COVID-19.

## 1. Introduction

Available data strongly suggest that introduction of COVID-19 vaccines was successful in controlling the COVID-19 pandemic and reduced SARS-CoV-2-related deaths [1,2]. Most of the COVID-19 vaccine development has so far focused on the response against the viral S (spike) protein, which by induction of neutralizing antibodies directed against the receptor-binding domain (RBD) of the S protein can protect against infection by preventing the virus from binding to its cellular receptor, ACE2 [3]. Currently, mainly genetic vaccines which deliver nucleic acid coding for the S protein with adenovirus vectors or introduce S-encoding mRNA into cells of the vaccinated person are licensed [4,5,6,7].

While these vaccines represented an initial breakthrough in the fight against COVID-19, several disadvantages of genetic vaccination have subsequently been recognized. Firstly, genetic vaccines targeting the S protein induce a quite variable immune response, possibly due to the fact that the amount of antigen produced by the host cells cannot be controlled, and a considerable percentage of vaccinated subjects, in particular vulnerable patients, remain non-responders [7,8,9,10,11]. Furthermore, genetic vaccination-induced antibody responses drop relatively quickly in titers [12,13,14], ensuring that repeated vaccinations are necessary to maintain protective immunity. Accordingly, adenovirus-based vaccines are now used less frequently because repeated boosting may reduce vaccine efficacy by inducing adenovirus-specific antibodies. Immunization with genetic SARS-CoV-2 vaccines, although generally considered safe, has also been associated with adverse events such as myocarditis [15,16,17], thromboembolic events [18,19,20], allergic reactions [21,22], neurological problems [23] and immunization-related deaths [24]. These associations are strengthened by detectable S antigen in blood combined with increased cytokine production being a notable marker of negative post-mRNA vaccine symptoms [25]. A potential solution to the side effects of genetic COVID-19 vaccines would be to change the platform and develop recombinant subunit vaccines, to be used in cases where genetic vaccines are unlikely to be safe or effective, or to provide long-lasting immunity in situations where arranging multiple boosting is difficult.

A more serious problem with S-based vaccines is the emergence of novel SARS-CoV-2 variants, in particular Omicron and its subvariants, which feature significantly different S sequences from the earlier strains against which the vaccines have been created. Both neutralizing antibody and T cell responses are affected by S protein alterations in the new variants [26,27], accordingly, it has become clear that currently available COVID-19 vaccines may confer less protection against Omicron infection [28,29,30]. One way to overcome the immune evasion of newly emerging SARS-CoV-2 variants is to consider conservative viral proteins as vaccine targets, for example the N (nucleocapsid), M (membrane) and E (envelope) [31,32,33,34]. Protein N, in particular, is an interesting vaccine candidate since it is one of the most conserved antigens [35,36,37] and is highly abundant [38,39]. T-cell immune responses specific for N protein of SARS-CoV-1 were found to be extremely long-lived (i.e., more than 17 years post infection) [40], and pre-clinical studies showed that specific anti-N antibodies against SARS-CoV-2 were maintained at a plateau for more than 5 months in mice after immunization [37]. A recent study has demonstrated that N can appear on the surface of infected cells, indicating that it can be a target for antibody-dependent cellular cytotoxicity (ADCC) and for cytotoxic T-cells [41]. Furthermore, several studies indicate that vaccination with N in combination with S or RBD may increase anti-SARS-CoV-2 immunity [42,43,44]. Accordingly, researchers have started to investigate N as a candidate antigen for the development of a SARS-CoV-2 vaccine [45,46,47].

Here we report the development, physicochemical, immunological and in vivo characterization of a SARS-CoV-2 subunit vaccine, Convacell^®^, which is based on recombinant N protein formulated as squalane-based emulsion. Importantly, we demonstrate that immunization with Convacell^®^ protects Syrian hamsters against development of severe disease after SARS-CoV-2 infection.

## 2. Materials and Methods

### 2.1. Production of N Protein and Vaccine Formulation

The nucleotide sequence of wild-type SARS-CoV-2 N protein (GenBank: YP_009724397.2) was optimized for expression in *E. coli* and cloned into the pET-28b(+) vector. *E. coli* strain Lemo21 (DE3) (NEB, Ipswich, United States, cat. C2528J) was transformed with the obtained plasmid. Culture media was grown in a 5 L Biostat B bioreactor (Sartorius, Göttingen, Germany) using LB media supplemented with 0.3% glycerol. Induction of protein expression was achieved using 5052 media [48]. Cells were harvested and stored at −80 °C. After thawing biomass was homogenized using Panda plus 1000 (GEA, Düsseldorf, Germany) in lysis buffer (50 mM Tris-HCl, pH 8, 1 M NaCl, 0.1% Triton X-100, 4 mM MgCl_2_, 5 mM EDTA, 0.1 mM PMSF) and the soluble fraction of proteins was collected for further purification. Host proteins were precipitated in 15% ammonium sulfate and removed. Then target protein fraction was precipitated in 30% ammonium sulfate and resuspended in buffer for ion-exchange chromatography (50 mM Tris-HCl buffer, pH 9, 0.1 mM PMSF). The first step of purification was anion-exchange chromatography (Capto Q by Cytiva, Marlborough, United States) in flow-through mode, followed by a cation-exchange step (WorkBeads 40S by BioWorks, Boston, United States) in binding mode and elution with 1 M NaCl gradient was carried out. The hydrophobic interaction chromatography (Capto Phenyl ImpRes by Cytiva, Marlborough, United States) in binding mode was performed after addition of ammonium sulfate to 1 M final concentration in phosphate buffer (50 mM sodium phosphate, pH 7.5, 5 mM EDTA). Elution was performed with 1 M to 0 M ammonium sulfate gradient. Finally, the buffer was exchanged to phosphate buffered saline (PBS) using 30 kDa MWCO tangential filtration (Vivaflow 200 by Sartorius, Göttingen, Germany). Protein N solution was sterile filtered through a 0.22 μm PES membrane (Sarstedt, Nümbrecht, Germany) and stored at 4 °C before use.

In all in vivo experiments 50 μg of protein per dose were injected into animals. In the case of a 0.5 mL dose volume the protein N solution in PBS was mixed in an equal volume of squalane-based emulsion (squalane, (D,L)-α-tocopherol, polysorbate 80) to obtain final concentration of 0.1 mg/mL. In the case of a 0.2 mL dose volume a final concentration of 0.5 mg/mL concentration of protein N solution was used. The emulsion was prepared by mixing squalane, (D,L)-α-tocopherol and polysorbate 80 in PBS and further sonicating. The vaccine formulation contained a final concentration of: 30 mg/mL squalane (Acros, Geel, Belgium, cat. 215355000), 10 mg/mL (D,L)-α-tocopherol (Sigma-Aldrich, St. Louis, United States, cat. 90669) and 10 mg/mL polysorbate 80 (Sigma-Aldrich, cat. 59924). The placebo preparations contained 30 mg/mL squalane (Acros, Geel, Belgium, cat. 215355000), 10 mg/mL (D,L)-α-tocopherol (Sigma-Aldrich, St. Louis, United States, cat. 90669) and 10 mg/mL polysorbate 80 (Sigma-Aldrich, St. Louis, United States, cat. 59924) and PBS instead of N protein.

### 2.2. Stability of Vaccine

The stability of the vaccine was assessed by measurement of N-specific antibodies that are induced by immunization with samples of vaccine stored at 2–8 °C for 0, 1, 2, 3, 6, 9, 12 and 15 months. Vaccines from three lots were used for study (Lot #490821R, #500821R and #560821R, SPbSRIVS, Saint-Petersburg, Russia). For each time point two groups of female Balb/c mice, 6–8 weeks age, 6 animals per group, were immunized with either placebo or Convacell^®^ on day 0 and 7. On day 14 blood was collected, processed to plasma and analyzed using an ELISA protocol described in “Measurement of N-specific antibodies” section of Materials and Methods.

### 2.3. N Protein Characterization

Electrophoresis: SDS-PAGE protein separation in 15% gel was performed using a Mini-Protean Tetra Cell setup (Bio-Rad, Hercules, United States) with 200 V voltage until dye reached the bottom of the gel. Native PAGE was run for 2 h using 6% gel and 200 V voltage with inverted polarity due to the positive charge of protein N. M_w_ marker from 10 to 250 kDa (NEB, Ipswich, United States, cat. P7719S) was used. Staining was performed with Coomassie R250 (Bio-Rad, Hercules, CA, USA).

Western blotting (WB): an SDS-PAGE of the protein sample was run on a 12.5% gel, with further transfer onto a 0.2 µm PVDF membrane (Bio-Rad, cat. 1620177) using a semi-dry transfer method (Towbin transfer buffer) [49]. After transfer, the membrane was incubated in blocking buffer (3% non-fat milk in 20 mM Tris-HCl pH 7.5, 500 mM NaCl, 0.05% Tween-20, 0.2% Triton-100) by placing it on an orbital shaker for 60 min. The membrane was further incubated for 60 min with anti-N antibodies (Anti-SARS-CoV-2 Nucleocapsid antibody, clone 1A6, Antibodies Online, Aachen, Germany, cat. ABIN6952664) diluted to a titer of 1:5000 with blocking buffer. Anti-N antibodies are the recombinant mouse ScFv of clone 1A6, with a human IgG1 Fc tag on the C-terminal, expressed in HEK293. The membrane was further washed in blocking buffer for 10 min. The washing procedure was repeated four times. Following the washing procedure, the membrane was incubated for 60 min with secondary antibodies (rabbit anti-human IgG horseradish peroxidase conjugated, Bio-Rad, Hercules, United States, cat. 1706515) diluted to a titer of 1:2000 with blocking buffer. The membrane was further washed in the washing buffer (2 mM Tris-HCl pH 7.5, 250 mM NaCl, 0.05% Tween-20, 0.2% Triton X-100) for 10 min. The washing procedure was repeated three times. After washing, the sample was visualized on membrane by incubation in 3 mL of stabilized TMB solution for 3–5 min.

Dynamic light scattering correlograms for protein and vaccine samples were measured using the Zetasizer Nano ZS system (Malvern Panalytical, Malvern, United Kingdom) at 173° angle in a plastic 1 cm cuvette. The wavelength was 630 nm. The built-in Zetasizer Default software (v. 7.13, Malvern Panalytical, Malvern, UK) processing protocol was used to obtain intensity weighted size-distribution.

MALDI-MS analysis was performed using a Bruker ultrafleXtreme MALDI-TOF/TOF mass spectrometer. The whole protein molecular weight was determined in samples that were zip-tip desalted in advance. Protein sequencing was done on samples subjected to digestion by Trypsin-EDTA Solution 10X (Sigma-Aldrich, St. Louis, MO, USA, cat. 59418C-100ML). The sample was analyzed by SDS-PAGE then protein bands were cut and washed twice with 30 mM ammonium bicarbonate, 40% acetonitrile in water. The remaining water was removed by incubation of the cut band in 100% acetonitrile and drying. The trypsin at concentration 20 μg/mL in 50 mM ammonium bicarbonate buffer was added to gel fragments. After incubation for 18 h at 37 °C the reaction was stopped by addition of 0.1% trifluoroacetic acid. After preparation, samples were mixed with the DHB matrix (Sigma-Aldrich, St. Louis, United States) in equal volumes, applied to a steel target, and measured in the reflex mode of detecting positive ions. At least 5000 laser pulses were applied to accumulate each spectrum. Protein identification was carried out using MASCOT with SwissProt database (https://www.uniprot.org, accessed on 11 August 2022) [50,51]. The mass determination accuracy was limited to 20 ppm. At least three samples were measured in each experiment.

Residual bacterial endotoxins, were determined using Fujifilm Wako Chemicals (Richmond, United States) WPEPK4-20015 kit, residual *E. coli* protein and DNA contents were determined using Cygnus Technologies (Southport, United States) F410 and D410T kits, respectively. Kits were used according to manufacturer instructions.

DSC analysis. Differential scanning calorimetry analysis of N protein samples was performed in triplicate on 0.2 mg/mL samples of recombinant protein N obtained via the method described above. Analysis was performed in closed 100 μL cells. Analysis was performed twice. Temperature was increased at a rate of 1 K/min.

### 2.4. Animals and Ethics

All study protocols were approved by the respective ethical committees at facilities where studies were carried out. Decision credentials are given in Appendix A.

Animals were assigned to groups based on block randomization, with individual animals first being randomly assigned to one of five blocks, and then each block being randomly assigned to one of the experimental groups or the placebo group. No exclusion criteria for animals were established. Technicians performing the experiment were blinded to the nature of each group during the course of the study, with group assignment data available only to the study coordinators. During the course of the experiment, mice were housed in groups of ten animals of the same sex, on wood chip bedding in plastic cages with wire lids. Feed prepared according to Russian GOST (“State standard”) 34566-2019 was deposited in recesses on the lids and purified water was provided in sipper bottles. *Callithrix jacchus* monkeys were housed in groups of two animals of the same sex, in wire cages equipped with wooden hides/nests and wooden branches and shelves for enrichment. Feed prepared in the laboratory was deposited into each cage twice per day. Purified water was provided in autoclavable sipper bottles. A humane endpoint was established during the experiment, where an animal that reached a certain number of points on the observable distress rubric would be euthanized. However, no animals approached the humane endpoint. Syrian hamsters were housed in groups of five animals of the same sex in conditions otherwise identical to those of mice.

During the infection challenge experiment, infected Syrian hamsters were housed in BSL-3 facilities in the M.P. Chumakov Federal Scientific Center for Research and Development of Immunobiological Drugs. Rabbits were housed in groups of two animals of the same sex, in wire cages equipped with sipper bottles and rest areas with sheet flooring. Feed prepared according to Russian GOST (“State standard”) 32897-2014 was deposited into feeder apertures in cage doors. Purified water was provided in sipper bottles. Human PBMC-engrafted mice were housed in individually ventilated polysulfone cages with HEPA filtered air at a density of up to five mice per cage. Laboratory-formulated feed and filtered water were freely provided.

### 2.5. Safety Assessment

Vaccine safety assessment was performed in accordance with ICH M3 and ICH S6 guidelines and included the repeated dose toxicity (RDT) study in mice and rabbits, with the experimental unit being an individual animal. Sample sizes were chosen to involve the smallest possible number of animals while still providing accurate results.

Five groups of ten male and ten female mice were used, with one such group being used as control through injection with placebo, two groups injected with 0.5 mL of vaccine and two groups injected with 1 mL of vaccine. The total number of animals used in the experiment was 100.

Five groups of eight male and eight female Soviet Chinchilla rabbits were also used, with one such group being used as control through injection with placebo, two groups being injected with 0.5 mL of vaccine and two groups injected with 1 mL of vaccine. The total number of animals used in the experiment was 40.

Animals were immunized on days 1, 8 and 15. Half of the animals were euthanized on day 17 and the remainder on day 43. Mice were euthanized using CO_2_ asphyxiation followed by exsanguination. Rabbits were euthanized using Zoletil 100 (Virbac, Carros, France) and Xylazine (Interchemie, Venray, The Netherlands) overdose followed by removal of vital organs. Clinical observations were carried out daily to assess the presence of any signs of intoxication or local inflammation or irritation.

Weighing was performed weekly. Hematological parameters of the blood collected during euthanasia were assessed via Mythic 18 Vet (Orphée, Geneva, Switzerland) hematological analyzer and included: erythrocyte count, hemoglobin level, hematocrit, thrombocyte and leucocyte count and lymphocyte, monocyte and granulocyte abundance. Biochemical parameters of the blood samples were analyzed via Random Access ‘A-25′ (BioSystems, Barcelona, Spain) analyzer using reagents from the same company. The parameters included alanine aminotransferase, aspartate aminotransferase, urea, total protein, creatinine, cholesterol, albumin, globulin, triglyceride, bilirubin, alkaline phosphatase and glucose levels.

Aorta, heart, trachea, lungs with bronchi, thymus, esophagus, stomach, small intestine, large intestine, pancreas, liver, spleen, kidneys, bladder, adrenal glands, testicles, ovaries, popliteal lymph nodes, thyroid gland, brain, skin with the femoral muscle of the right and left pelvic limbs (injection site) were collected after euthanasia and examined via histological analysis. The RDT studies included assessment of electrocardiogram parameters, blood pressure, respiratory rate in rabbits and an open field test in mice that assessed horizontal locomotor activity, rearing, urination and defecation, shelter exploration and autogrooming. Pyrogenicity testing was carried out in rabbits (n = 3, experimental unit is a single animal) according to the Russian Pharmacopoeia [52].

### 2.6. Measurement of N-Specific Antibodies

N-specific IgG antibody responses were evaluated in Balb/c mice, Syrian hamsters and rabbits, with the experimental unit being an individual animal in each case. Sample sizes were chosen to involve the smallest possible number of animals while still providing accurate results.

One group of 40 female Balb/c mice, 6–8 weeks of age, was immunized intramuscularly with 0.5 mL of Convacell^®^ (SPbSRIVS, Saint-Petersburg, Russia), containing 50 μg of protein N, on days 0 and 14 of the experiment. An identical group of 40 mice was immunized with the same volume of placebo using the same schedule and used as the control. The total number of animals used in the experiment was 80.

On days 0, 7, 14, 21, 28, 43, 161 and 312, five animals from each group were anesthetized by 20 mg/kg Zoletil 100 (Virbac, Carros, France) and 5 mg/kg Xylazine (Interchemie, Venray, Netherlands) and blood samples were collected in tubes with 2 mg/mL EDTA. After that, the animals were subjected to exsanguination from the cavities of the heart. This type of euthanasia of animals is accompanied by a minimum of pain, suffering and distress and was carried out by competent staff. Additionally, mouse spleens were collected.

One group of 60 female Syrian hamsters, 6–8 weeks of age, was immunized intramuscularly with 0.5 mL of Convacell^®^ (SPbSRIVS, Saint-Petersburg, Russia) on days 0 and 14 of the experiment. An identical group was immunized with placebo with the same schedule and used as the control. The total number of animals used in the experiment was 120.

On days 0, 7, 14, 21, 28, 42, 161 and 386, five animals from each group were anesthetized using 20 mg/kg Zoletil 100 (Virbac, Carros, France) and 5 mg/kg Xylazine (Interchemie, Venray, Netherlands) and blood samples were collected in tubes with 2 mg/mL EDTA. After that, the animals were subjected to exsanguination from the cavities of the heart. This type of euthanasia of animals is accompanied by a minimum of pain, suffering and distress and was carried out by competent staff.

Two groups of ten male rabbits, 12–20 weeks of age, were immunized intramuscularly with 0.5 mL of either Convacell^®^ (SPbSRIVS, Saint-Petersburg, Russia) or placebo on days 0 and 14 of the experiment.

Blood was collected without euthanasia via catheter from the marginal dorsal ear vein on days 0, 7, 14, 21, 28, 42, 161 and 386.

After coagulation, blood samples were centrifuged at room temperature at 300× *g* for 15 min. Plasma was frozen at −80 °C for further analysis.

The IgG N protein titer specific for SARS-CoV-2 was determined by indirect enzyme immunoassay (ELISA). Ninety-six-well plates with a high degree of binding were used for analysis (Costar by Corning, Corning, United States, cat. 2592). The plates were coated with N-protein (SPbSRIVS, Saint-Petersburg, Russia) at a concentration of 1 μg/mL in 0.05 M carbonate–bicarbonate buffer solution (pH 9.6) 100 μL/well and incubated for 1 h at 37 °C and 600 rpm on a ST-3L shaker (ELMI, Riga, Latvia). The wells were washed with 300 μL of a washing buffer solution (0.01 M PBS containing 0.1% polysorbate 20, pH 7.4). To block non-specific binding sites, 300 μL of blocking buffer solution (0.5% milk powder in the washing buffer) was added to the wells and incubated for 30 min at 37 °C 600 rpm. At the end of incubation, the wells were washed four times with a washing buffer solution. Before analysis, mouse serum samples, positive and negative controls were diluted in blocking buffer solution. Serial dilutions of mouse samples were prepared 1:250–256,000 times. A 100 μL blank (in at least 10 replicates), samples of positive and negative controls (in 2 replicas), serial dilutions of the studied samples (in 3 replicas) were added to the wells of the tablet. A blocking buffer solution was used as a blank for the analysis of samples. The sample plates were incubated for 1 h at 37 °C 600 rpm. At the end of incubation, the wells were washed four times with a washing buffer solution. At the next stage, 100 μL of diluted (1:64,000) secondary antibodies according to the species (HPR Goat Anti-Mouse IgG (H + L) (cat. 170-6516, Bio-Rad, Hercules, United States), HRP Rabbit Anti-Hamster IgG (H + L) (cat. SAB3700491, Sigma-Aldrich, St. Louis, United States), HRP Goat Anti-Rabbit IgG (H + L) (cat. ab6721, Abcam, Cambridge, United Kingdom) were added to each well and incubated for 1 h at 37 °C 600 rpm, washed four times with 300 μL of washing buffer solution. Then, 100 μL of a solution of 3,3′,5,5′-tetramethylbenzidine substrate (TMB-substrate, JSC BTK Bioservice, Moscow, Russia) was added to the wells, the plate was incubated in the dark at room temperature for 12 min. To stop the reaction, 100 μL of a stop solution (0.18 M sulfuric acid) was added to each of the wells. The optical density was measured using a CLARIOstar (BMG Labtech, Ortenberg, Germany) multifunctional microplate reader at two wavelengths of 450 nm (main wavelength) and 620 nm (control wavelength). Calculations were carried out for the optical density at 450 nm. The mean optical density for the blank wells (DBlk) was calculated. A reference wavelength of 620 nm was used to assess the degree of reaction termination. The positive–negative threshold (cut-off point) was calculated using the formula: Cut-off = DBlk + 0.05. The cutoff value was subtracted from the optical density of the samples. The dilution of the sample was taken as a titer when the optical density value after subtracting the Cut-off value was positive and the smallest obtained for the entire series of dilutions.

The specificity of the ELISA method for the N-protein of SARS-CoV-2 was evaluated for different manufacturers of protein N used for immobilization on plates. Measured anti-N antibody titers were the same within one dilution for N protein produced by SPbSRIVS (Saint-Petersburg, Russia), Biovendor (Brno, Czechia, cat. RI973598100) and Saint-Petersburg Pasteur Institute (Saint-Petersburg, Russia, cat. N-CoV-2-IgG PS).

### 2.7. Measurement of N-Specific Cellular Responses

Two groups of four marmoset monkeys (*Callithrix jacchus*) each, 2–5 years old, were used to assess N-specific lymphocyte proliferation, with the experimental unit being an individual animal. Animals from one group were injected intramuscularly with 0.5 mL of vaccine preparation (50 μg of protein N) on days 10 and 31. Animals from the other group were injected in the same manner with placebo.

On days 0, 10, 24, 38 and 45, aliquots of 1 mL of blood were taken from the animals by puncture of the femoral vein using syringes with a volume of 2.5 mL with 25 G needles, in which 0.1 M sodium-EDTA solution was pre-added (the final concentration is 5 mM). Peripheral blood mononuclear cells were isolated on lymphocyte isolation medium (LSM; MP biomedicals, Solon, USA) according to the manufacturer’s protocol. Cells in the resulting suspension were counted in a Goryaev chamber via a Zeiss light microscope Observer A1 (Zeiss, Oberkochen, Germany). Cells were then stained with CFSE (CellTrace, Invitrogen, Waltham, United States) via a 10-min incubation. Aliquots of 2 × 10^5^ PBMCs were suspended in 200 μL of complete medium RPMI-1640 (PanEco, Moscow, Russia) supplemented with 10% FBS (Hyclone, Logan, USA) and penicillin-streptomycin mixtures (PanEco, Moscow, Russia) were added into wells of a 96-well plate and stimulated with 1 µg/mL of peptivator (PepTivator^®^ SARS-CoV-2 ProtN, Miltenyi, Bergisch Gladbach, Germany, cat. 130-126-698). Five µg/mL of concanavalin A (PanEco, Moscow, Russia) was used as positive control. The cells were incubated for 96 h in 5% CO_2_ at 37 °C, then stained for surface antigens CD3, CD4 and CD8 via purified anti-marmoset antibodies (BD, Biolegend, San Diego, United States) and viability via 4′,6-diamidino-2-phenylindole (DAPI). Proliferation detection was carried out by flow cytometry on the CytoFlex device (Beckman Coulter, Brea, United States) with 10^4^ events per gate. Specific proliferation was considered to occur if more than 0.5% of cells in subpopulation were detected with decreased CFSE fluorescence intensity.

Harvested Balb/c mouse spleens were used to obtain splenocytes through homogenization and filtration through cell strainers. The number of cells in the cultures was determined using a Countess cell counter (Invitrogen, Waltham, United States). Two million cells were seeded in 24-well culture plates in a total volume of 1 mL of complete culture medium (RPMI-1640 with 5% fetal bovine serum, 2 mM L-glutamine, 100 U/mL penicillin, 100 μg/mL streptomycin) per well. Three wells with cells were prepared. In the first well, peptides from the N protein were added according to manufacturer instructions (PepTivator^®^ SARS-CoV-2 ProtN, Miltenyi, Bergisch Gladbach, Germany, cat. 130-126-698). In the second well, concanavalin A was added to the cells at 5 μg/mL final concentration as a positive control. In the third well the equal volume of RPMI-1640 medium was added as a negative control. Cells were cultured for 96 h in 5% CO_2_ at 37 °C. After incubation, the plates were agitated on a shaker (5 min, 300 rpm). Then the contents of the well were completely transferred into microtubes. The samples were centrifuged to pellet the cells (5 min, 300 g). Aliquots of the supernatant were taken and frozen at −70 °C. A multiplex immunofluorescent assay for murine cytokine quantification was performed using a High Sensitivity T-Cell Magnetic Bead Panel kit (Merck, Burlington, United States, cat. MHSTCMAG-70K) on Bio-Plex 200 (Bio-Rad, Hercules, United States,) according to the manufacturer’s protocol. Results were counted if cytokine concentration was above the one determined in the negative control and below the one in the positive control.

Human PBMC-engrafted NSG-(KbDb)^null^ (IA)^null^ mice (JAX Stock #025216) were used to study human chemokine and cytokine levels in murine blood after vaccination with Convacell^®^ (SPbSRIVS, Saint-Petersburg, Russia), with the experimental unit being an individual animal. Mice 7–8 weeks old were divided into four groups of ten animals each: group 1—intact animals, group 2—PBMC-engrafted animals, group 3—PBMC-engrafted and injected with placebo, and group 4—PBMC-engrafted and injected with vaccine. On day 0, mice in groups 2–4 were irradiated with 140 cGy from an X-ray irradiator source. At least 4 h post-irradiation, 10^7^ PBMCs from a single COVID-19 naïve donor per mouse were intravenously injected in 100 μL volume in PBS. Mice in groups 3 and 4 on day 5 and day 19 were injected with placebo (formulation without protein N) or vaccine intramuscularly with 0.1 mL vaccine preparation (50 μg of protein N), respectively. During the study one animal in group 2, one animal in group 3 and two animals in group 4 were discarded due to premature death. On study day 35 all animals were euthanized via CO_2_ asphyxiation. Blood was collected by cardiac puncture, processed to plasma and sent for analysis of human cytokines (V-PLEX Human Cytokine 30-Plex Kit, by Meso Scale Discovery, Rockville, United States, cat. K15054D-1).

### 2.8. SARS-CoV-2 Infection Challenge of Syrian Hamsters

Female Syrian hamsters, 6–8 weeks old, were divided into four groups of ten animals each: group 1—intact animals, group 2—infected animals, group 3—injected with placebo and infected, and group 4—injected with Convacell^®^ (SPbSRIVS, Saint-Petersburg, Russia) and infected. Animals in groups 3 and 4 on day 0 and day 14 were injected with placebo (formulation without protein N) or with the vaccine intramuscularly with 0.5 mL vaccine preparation (50 μg of protein N), respectively. The experimental unit was an individual animal. The total number of animals used in the experiment was 40.

On day 29, anesthetized animals of groups 2–4 were infected intranasally with 10^5^ TCID_50_ of SARS-CoV-2 in 50 μL of buffer (Clinical isolate strain PIK35 GISAID ID EPI_ISL_428852) [53]. After infection (i.e., day 29), body weights of all animals were monitored daily. On day 3 after infection five animals from each group were euthanized by exsanguination under 20 mg/kg Zoletil 100 (Virbac, Carros, France) and 5 mg/kg Xylazine (Interchemie, Venray, Netherlands) anesthesia. Animal lungs were then harvested. Lungs were weighted to calculate the % of body weight. The left lung was then used for histopathological evaluation in RMC “Home of pharmacy” JSC, the right lung was used for RT-PCR analysis. On day 7 after infection the same procedures were carried out with remaining five animals in each group.

The lungs were homogenized using a TissueLyser homogenizer (Qiagen, Hilden, Germany). The obtained homogenate was analyzed using a Polyvir SARS-CoV-2 assay (Lytech, Moscow, Russia) according to the manufacturer’s instructions with the determination of Ct (threshold cycle of fluorescence). In detail, 200 µL of homogenate and 5 µL of internal control samples, were added to the RNA express reagent, vortexed for 10 s, incubated for 15 min at +95 °C and centrifuged for 1 min at 12,000 rpm. After that, 7 µL of the obtained supernatant was used for setting up a RT-PCR. A total of 28 µL of the amplification mixture (buffer solution containing enzymes, primers, probes, nucleotides, etc.) was added to the reaction. RT-PCR program: +42 °C-40 min; +95 °C-2 min; +95 °C-20 s; +64 °C-40 s-5 cycles; +95 °C-20 s; +64 °C-40 s (signal reading)-40 cycles. Ct was determined according to the instructions. Samples with an undetectable signal (Ct > 40) were considered negative.

The material for histological examination was fixed in a 10% neutral formaldehyde solution for 24 h, then embedded in paraffin. Paraffin sections 5–7 μm in thickness were prepared and stained with hematoxylin and eosin. Analysis was performed by a certified veterinary pathologist who was blinded regarding the animal’s manipulations in an organization different from the one where animals were housed and manipulated. Morphological examination of the slides was performed using an Axio Scope A1 light optical microscope (Carl Zeiss Microscopy GmbH, Oberkochen, Germany). Microphotography was performed using an AxioCamICc 1 digital camera and ToupView software (v.4.11, ToupTek, Hangzhou, China). The severity of lung pathologic lesions were analyzed using the inflammation scoring system adapted from Osterrieder et al. [54] (see Appendix A).

### 2.9. Statistical Analysis

Group data was analyzed for normal distribution via the Shapiro–Wilk W test. If data was normally distributed, mean and SD were calculated, otherwise, median and interquartile range were calculated. Pairwise group comparisons were performed using a two-tailed unpaired Student’s T-test or Mann–Whitney U test [55]. Analysis was made using GraphPad Prism v.9.1.1 software (GraphPad, San Diego, CA, USA) and Statistica v.10.0 software (StatSoft, Tulsa, United States). Differences were considered significant at *p* < 0.05.

## 3. Results

### 3.1. Production, Formulation and Biochemical Characterization of a Subunit Vaccine Based on Recombinant N, Termed Convacell^®^

A dose (0.5 mL) of Convacell^®^ comprises 50 μg of recombinant protein N. Recombinant protein N was obtained by expression in *E. coli* in the soluble cell fraction. Purification of protein yielded approximately 40–50 mg of pharmaceutical quality grade protein per 1 l of culture medium. The final solution contained <100 ppm of residual *E. coli* protein, <200 ppm of host cell DNA and <10 units of endotoxin per ml. The migration of recombinant N in SDS-PAGE was in agreement with its predicted molecular weight (45.6 kDa) (Figure 1A). The Western blot (WB) probed with monoclonal anti-N antibodies revealed a main band at the same molecular weight as the recombinant protein in SDS-PAGE (see lane 2 in Figure 1A). Comparison of lanes 1 and 2 in Figure 1A reveals minor bands below the main protein N band—these minor bands were due to shortened forms of the N protein. MALDI-MS analysis identified them as C-terminally cleaved fragments (data not shown). The molecular weight and sequence of the recombinant N protein was confirmed using MALDI-MS (Figure 1B) with a sequence coverage of 96.7%. Underlined peptides were additionally evaluated using MS/MS fragmentation to confirm a lack of oxidative modifications and deamidation. The recombinant N protein forms oligomers that disassemble at 93 °C according to DSC data (Figure 1C). The distribution of the recombinant N protein under native conditions was assessed using native PAGE (lane 3 in Figure 1A) and dynamic light scattering (blue curve in Figure 1D), revealing that the protein exists as one species of 13 nm diameter corresponding to a tetramer.

The vaccine is formulated as a squalane-based emulsion stabilized with (D,L)-α-tocopherol and polysorbate 80. It provides antigen presentation in the form of particles with a mean diameter of approximately 150 nm (Figure 1D). It can be stored at 2–8 °C for at least 1 year without loss of specific activity (see Appendix A). The placebo preparation consisted of adjuvant and buffer but without N protein.

### 3.2. Convacell^®^ Induces a Robust and Long-Lasting Antibody Response in Mice, Rabbits and Syrian Hamsters

Convacell^®^ was evaluated for its ability to induce N-specific antibodies by immunization of mice, rabbits and Syrian hamsters. Vaccine administration led to production of N protein specific IgG already at day 14 in all three species studied (Figure 2). The value of antibody titers peaked on days 21–28 after the first immunization. Importantly, antibody responses remain high even on day 386 indicating that Convacell^®^ induces a sustainable N-specific antibody response as was found in infected subjects and in preclinical studies with N-based vaccines [36,37,56,57]. The specificity of vaccine-induced anti-N antibodies and their sustained levels were also confirmed in clinical trials with Convacell^®^ (phase 1/2) [58] using Abbott Architect anti-N IgG detection system.

### 3.3. Convacell^®^ Induced N-Specific T Cell Response with Mixed Th1/Th2 Phenotype

In the first set of experiments, we studied if immunization with Convacell^®^ can induce N-specific T cell responses in *Callithrix jacchus* marmoset monkeys. The percentages of N-specific proliferated CD3^+^, CD4^+^ and CD8^+^ T cells were determined in immunized animals and in non-immunized animals via a CFSE dilution technique [59] (for FACS gating strategy, see Appendix A) at different time points after vaccination. In the vaccinated group, CD3^+^, CD4^+^ and CD8^+^ cells subpopulations proliferate after stimulation with N-peptivator in immunized animals (Table 1). The vaccinated animals had circulating N-specific CD3^+^, CD4^+^ and/or CD8^+^ cells on the 14th and 28th days post-vaccination, though none were detected on the 35th day post-vaccination. No T cell proliferation of any of the cell types at any time was detected in the placebo group which had received the emulsion formulation without recombinant protein N.

After we showed that immunization with Convacell^®^ can induce N-specific CD4^+^ and CD8^+^ T cell responses, we investigated the type of the N-specific cytokine response. We used two experimental animal models to study this question: Balb/c mice and NSG mice which had been engrafted with PBMC from a COVID-naïve, unvaccinated human subject.

Figure 3 shows that immunization with Convacell^®^ induced a mixed Th1/Th2 cytokine secretion profile in mice which was characterized by increased N-specific IFNγ and IL-4 levels in cultured PBMC as compared to non-immunized mice. A similar observation was made for immunized NSG mice—immunization with Convacell^®^ also induced a mixed Th1/Th2 human cytokine secretion profile as indicated by elevated N-specific human IFNγ and IP-10 levels (i.e., Th1) and elevated human IL-4 and IL-5 levels (i.e., Th2) in the plasma of immunized mice as compared to non-immunized mice (Figure 3).

### 3.4. Vaccination with Convacell^®^ Is Safe and Protects Syrian Hamsters against SARS-CoV-2 Infection

In order to assess the safety of Convacell^®^, we performed an extensive repeated dose toxicity study in mice and rabbits with different doses of Convacell^®^: the anticipated human dose and the double human dose. The results of the repeated dose toxicity studies demonstrated that Convacell^®^ is safe and well-tolerated and does not affect any organ systems, blood or laboratory parameters compared to placebo. A detailed presentation of these results can be found in the supplementary part of this article reporting the hematological and biochemical blood analysis, clinical observations, weighting, heart and circulatory parameters, neurological assessment, electrocardiogram, pyrogenicity and other relevant safety data (Appendix A). The only adverse effects found for Convacell^®^ were local irritations at the site of injection, which could be attributed to the use of the squalane-based oil in water adjuvant [60].

Results obtained in a Syrian hamster SARS-CoV-2 infection model demonstrate that Convacell^®^-vaccinated and then infected hamsters increased their body weight significantly earlier as compared to non-vaccinated infected animals on day 6 after infection (Figure 4A). Furthermore, there was approximately one order of magnitude reduction in virus RNA concentration in vaccinated animals compared to non-vaccinated or placebo-treated animals in the lower respiratory tract 7 days after challenge as determined by RT-PCR (Figure 4B). The significant increase in lung mass coefficient induced by infection is not observed in the vaccinated + infected group as opposed to placebo-treated + infected and non-treated + infected groups (Figure 4C). Histological analysis of lungs on day 7 after challenge revealed a tendency (*p* = 0.1) for a decrease in overall score for inflammation, edema and infiltration in vaccinated animals compared to the placebo group. Interestingly, a reduction in the lung histology score was also observed by placebo treatment containing only the adjuvant. However, vaccination led to a significant (*p* = 0.02) decrease in histopathology score compared to the negative control group (Figure 4D). Histological changes were well correlated with decrease in macroscopic signs of pneumonia in the experimental group. Figure 5 and Appendix A illustrate the reduced lung inflammation in the vaccinated group.

## 4. Discussion

Several lines of evidence indicate that the nucleocapsid (N) protein of SARS-CoV-2 may represent a possible target for vaccination against COVID-19. Vaccines targeting the N protein of the virus have been suggested as an intervention scenario alternative to approached targeting of the RBD-ACE2 binding [36,37,42,43,44,45,46,61,62]. In particular, N protein is capable of inducing the creation of tissue memory cells (T_RM_ cells) in lungs, which for years retain the ability to respond to the pathogen, stopping its spread at the very beginning of the infectious process [47]. Furthermore, it has been shown that immunization with N proteins from other viruses such as the influenza A virus protects from lethal infection with various strains [63,64]. Likewise, vaccination with N-protein from Dengue virus has been suggested [65].

Along these lines, our study reports the development, biochemical, immunological and in vivo characterization of a subunit vaccine for COVID-19 (i.e., Convacell^®^) which is based on *E. coli*-expressed purified N protein. Protein N is formulated as a squalane-based emulsion. The vaccine is stable for at least 1 year at 2–8 °C. Repeated dose–toxicity studies performed in two animal models demonstrated that Convacell^®^ is a safe and well tolerated vaccine. It induced a robust, early rising (i.e., already 14 days after the first injection) and long-lasting N-specific antibody response. Convacell^®^ stimulated an N-specific CD4^+^ and CD8^+^ T cell response which was characterized by a mixed Th1/Th2 cytokine profile in mice and in NSG mice engrafted with human PBMCs similar to that observed after natural SARS-CoV-2 infection [66]. Importantly, vaccination with Convacell^®^ protected Syrian hamsters against infection with SARS-CoV-2 as demonstrated by reduced weight loss, lowered virus RNA and ameliorated lung pathology as compared to non-immunized animals. Interestingly, a significant reduction in virus replication as compared to placebo, or no vaccination, was only noted for the N protein-containing vaccine whereas regarding lung histology/inflammation adjuvant alone seemed to also exert some positive effect, indicating that administration of squalene has a positive effect presumably by activation of innate immunity against SARS-CoV-2.

Not being based on the RBD domain of the S-protein, Convacell^®^ clearly has other modes of action as compared to S-protein or RBD-based vaccines. According to our data and current knowledge of SARS-CoV-2 biology [41] the following mechanisms may be considered for the protective effect of Convacell^®^. First, vaccination with Convacell^®^ induces IgG antibodies against N which may recognize SARS-CoV-2 infected cells presenting N on their surface as has been recently demonstrated by López-Muñoz et al. [41], and through mechanisms of ADCC may contribute to the elimination of infected cells. In this context it should be mentioned that internal virus proteins have been demonstrated to be presented on the surface of infected cells, for influenza A [67], vesicular stomatitis virus [68], measles [69], respiratory syncytial virus [70], lymphocytic choriomeningitis [71] and human immunodeficiency virus [72].

We also observed the induction of N-specific CD8^+^ T cell responses after vaccination with Convacell^®^. Cytotoxic CD8^+^ T cells may also contribute to the elimination of virus-infected cells expressing N protein. Furthermore, it is possible that N-specific antibodies can enter the infected cells which are susceptible to antibody uptake. N-specific antibodies may then interfere with virus replication. In fact, it has been shown that cells may be either intrinsically or transiently permissive for antibody penetration [73,74], which would enable interactions of intracellular N protein and anti-N antibodies. This could hinder N protein interaction with nucleic acids or its dimerization, which would hamper virion assembly and viral RNA replication [75]. At present we can only speculate about the mechanisms by which vaccination with Convacell^®^ protects against development of severe disease after SARS-CoV-2 infection because experiments in this direction would be beyond the scope of our study. However, the overall strong immunogenicity and results obtained by using the infection model of Syrian hamsters demonstrate the in vivo efficacy of Convacell^®^.

It should also be mentioned that several other studies support the usefulness of vaccination with N-based SARS-CoV-2 vaccines. For example, there are studies suggesting that vaccination with N and S or RBD may increase anti-SARS-CoV-2 immunity [42,43,44]. It may therefore be envisioned that Convacell^®^ can be combined with other S- or RBD-targeting vaccines to increase the armamentarium of COVID-19 vaccines. Possible advantages of N-based vaccines could be that the N-protein is highly conserved among different coronavirus strains and that accordingly, one may expect broader cross-protection to emerging virus variants. In fact, the N protein of SARS-CoV-2 has 90% homology with SARS-CoV-1, 99% with Bat-CoV, 88% with Pangolin CoV [76]. It is also likely, that there will be fewer non-responders to N-based vaccines as compared to S-based vaccines.

In summary, Convacell^®^ is an effective, highly stable under regular storage conditions and easy to manufacture and transport SARS-CoV-2 vaccine. The usage of conservative N protein as an antigen allows Convacell^®^ to be considered as a candidate pan-sarbecovirus vaccine. It may be useful in the fight against the COVID-19 pandemic despite the emergence of new virus variants [77], and accordingly clinical trials with Convacell^®^ have been initiated [58].

## Figures and Tables

**Figure 1 vaccines-11-00874-f001:**
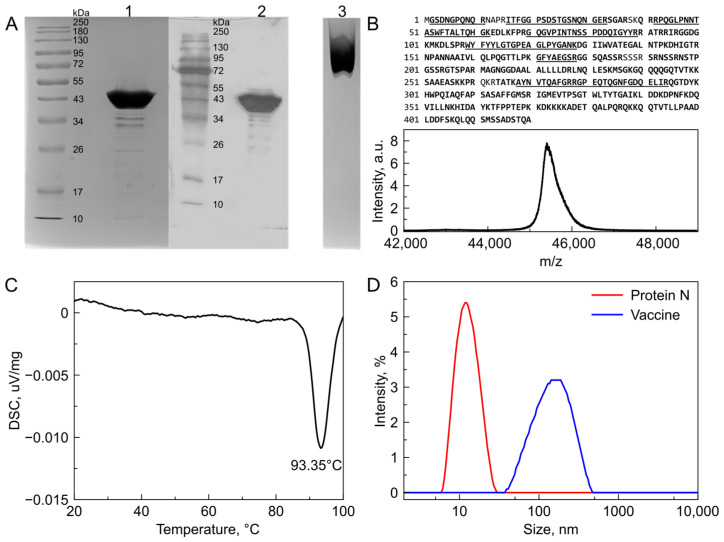
Biochemical characterization of recombinant SARS-CoV-2 N protein. (**A**) SDS-PAGE and Western blotting confirm the molecular weight and recognition by N-specific antibodies. Native PAGE shows homogeneous distribution of recombinant N as one species. Lanes: 1—SDS-PAGE of recombinant N protein solution; 2—Western blot of recombinant N protein solution; 3—native PAGE of recombinant N-protein solution. Uncropped photo of Western Blot membrane is presented in Appendix A. (**B**) MALDI-MS confirms the predicted molecular weight of monomeric recombinant nucleocapsid protein and 96.7% of sequence coverage was achieved. Confirmed sequence is marked in bold. Peptides subjected to MS/MS analysis are underlined. (**C**) Differential scanning calorimetry thermograms of 0.2 mg/mL N-protein samples show no endothermic peaks corresponding to protein globule melting but show a distinct exothermic peak at 93 °C corresponding to oligomer particles melting. (**D**) Nucleocapsid protein forms oligomeric structures with a hydrodynamic diameter of approximately 13 nm (red line) which corresponds to a M_w_ of approximately 200 kDa suggesting the formation of a tetramer. The emulsion formed by squalane, (D,L)-α-tocopherol and polysorbate 80 exposes the protein on oil in water drops with a size of approximately 150 nm in diameter (blue line).

**Figure 2 vaccines-11-00874-f002:**
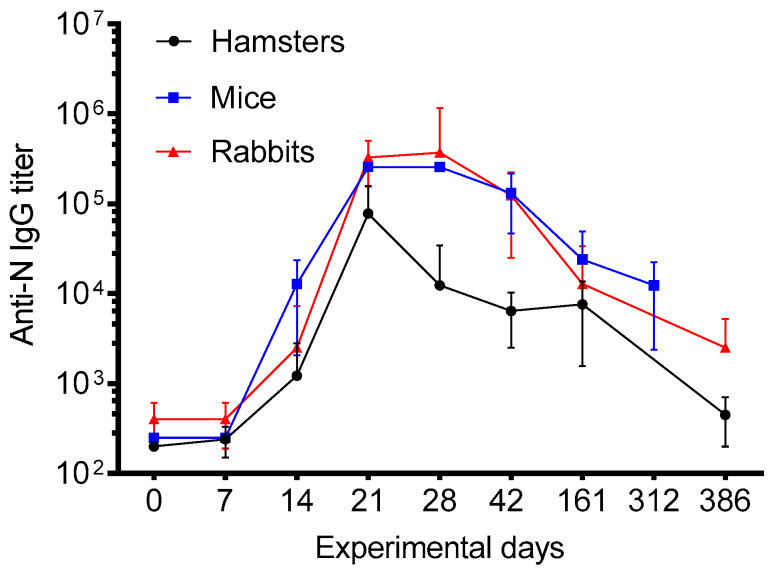
N-specific IgG titers (*y*-axis: median values and Q1:Q3 ranges) measured in immunized mice (n = 5 per time point), hamsters (n = 5 per time point) and rabbits (n = 10 per time point). Animals were immunized on days 0 and 14.

**Figure 3 vaccines-11-00874-f003:**
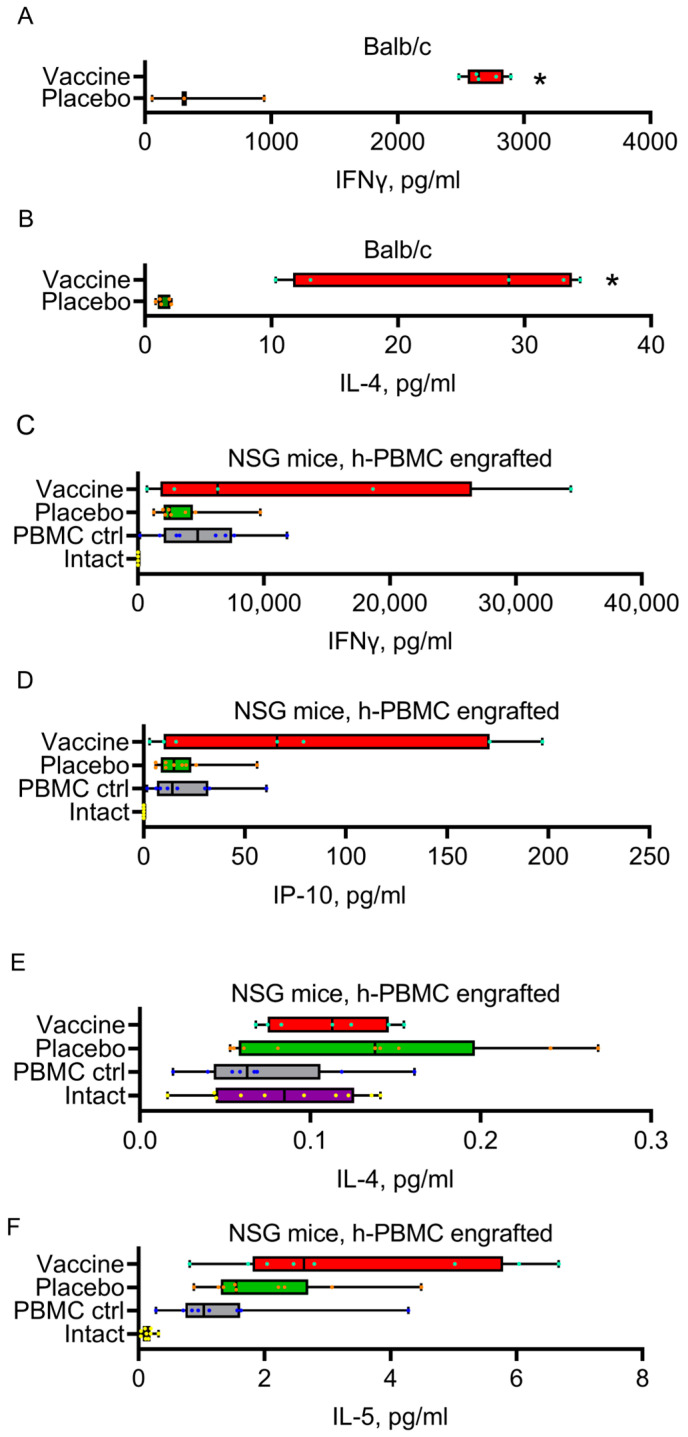
Murine IFNγ (**A**) and IL-4 (**B**) levels in supernatants of splenocytes isolated from Convacell^®^-immunized or from non-immunized Balb/c. Human IFNγ (**C**), IP-10 (**D**), IL-4 (**E**) and IL-5 (**F**) levels in plasma from Convacell^®^-immunized or from non-immunized NSG mice engrafted with human PMBCs. On panels C-F “PBMC ctrl” represents PBMC-engrafted NSG mice, “Intact” represents non-engrafted NSG mice. Statistically significant differences (*p* < 0.05) according to the Mann–Whitney test with two-stage step-up are indicated with an asterisk.

**Figure 4 vaccines-11-00874-f004:**
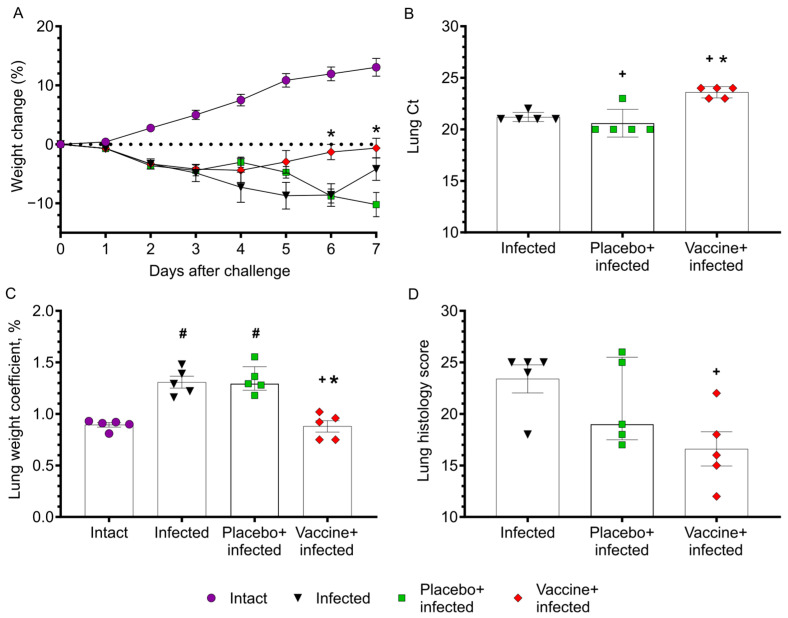
Protective effects of Convacell^®^ in a Syrian hamster model. (**A**) Hamster body weight changes after SARS-CoV-2 challenge, M ± SEM. (**B**) Virus RNA quantification results presented as a Ct RT-PCR parameter, M ± SEM. (**C**) Lung weight coefficient, M ± SEM. (**D**) Sum of lung histology score (see Appendix A for details of score rubric), Me(Q1;Q3). Statistically significant differences (*p* < 0.05) according to the Mann–Whitney test with two-stage step-up are indicated: ‘#’ when compared to Intact group; ‘+’ when compared to Infected group; ‘*’ when compared to placebo-treated + infected group. More details of statistical analysis can be found in Appendix A.

**Figure 5 vaccines-11-00874-f005:**
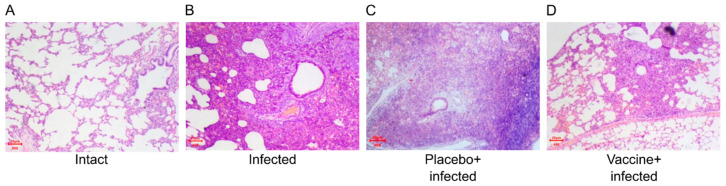
Representative examples of microscopic (magnification 40×) lesions of SARS-CoV-2 in Syrian hamsters. (**A**) Intact animals’ group, (**B**) non-vaccinated and infected group, (**C**) placebo-treated and infected group, and (**D**) vaccinated and infected group. See also Appendix A.

**Table 1 vaccines-11-00874-t001:** Percentages of proliferated CD3^+^, CD4^+^ and CD8^+^ T cells in immunized marmoset monkeys (n = 4; M = male; F = female) at different time points after vaccination. Values < 0.5% are considered as being below the cut-off.

Animal, Sex	Cell Phenotype	Days after First Immunization			
		0	14	28	35
1, F	CD3^+^	-	0.82	-	-
	CD3^+^CD4^+^CD8^−^	-	0.63	-	-
	CD3^+^CD4^−^CD8^+^	-	0.54	0.55	-
2, M	CD3^+^	-	-	0.57	-
	CD3^+^CD4^+^CD8^−^	-	-	-	-
	CD3^+^CD4^−^CD8^+^	-	-	-	-
3, M	CD3^+^	-	0.76	0.90	-
	CD3^+^CD4^+^CD8^−^	-	1.07	0.81	-
	CD3^+^CD4^−^CD8^+^	-	-	0.52	-
4, F	CD3^+^	-	0.71	-	-
	CD3^+^CD4^+^CD8^−^	-	-	-	-
	CD3^+^CD4^−^CD8^+^	-	1.09	-	-

## Data Availability

All data will be made available by the authors upon request without reservation.

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
