# Peer review of "Immunogenicity and In Vivo Protective Effects of Recombinant Nucleocapsid-Based SARS-CoV-2 Vaccine Convacell®"

_vaccines, 2023, doi:10.3390/vaccines11040874_

Round 1

Reviewer 1 Report

This is an interesting paper about Immunogenicity and in vivo protective effects of recombinant nucleocapsid-based SARS-CoV-2 vaccine Convacell®. I suggest it for publication after the following points are solved.

1. Line 43-46, one recent review (ACS Nano  2021, 15, 11, 16982-17015) should be included to support such a claim.

2. The authors should discuss why such an emulsion without any positive charged molecules could load mRNA.

3. A section of conclusion and perspective is preferred.

Minor editing of English language required

Author Response

Response to Reviewer 1 Comments

Point 1: Line 43-46, one recent review (ACS Nano  2021, 15, 11, 16982-17015) should be included to support such a claim.  

Response 1: The claim made in the sentence in lines 43-46 lays in the field of vaccine platforms and antigen selection. At the same time the highly cited review (ACS Nano  2021, 15, 11, 16982-17015) describes advances in the field of delivery of mRNA in lipid nanoparticles. These two fields are adjacent and are in close contact; however, the paper suggested by reviewer does not support the main points of the claim: the prevalence of genetic vaccines and their choice of S-protein as an antigen. Therefore, we believe that suggested paper is not a good fit for referencing in the sentence in lines 43-46.

Point 2: The authors should discuss why such an emulsion without any positive charged molecules could load mRNA.

Response 2: The topic of mRNA vaccine delivery in emulsions is undoubtedly one of the points that should be discussed if the vaccine contains mRNA as an antigen-encoding vector. The Convacell® vaccine contains: recombinant protein N, phosphate-buffered saline, squalane, (D,L)-α-tocopherol and polysorbate 80. mRNA or any other nucleic acids are not included in the composition. Moreover, the quantity of residual host cell DNA is confirmed to be below limits allowed for recombinant proteins by the US Pharmacopoeia. The squalane-based emulsion in the Convacell® vaccine is intended to carry a protein antigen, but not mRNA. Therefore, the discussion of how such an emulsion without any positively charged molecules could load mRNA will not be a good fit the discussion session in this paper.

Point 3: A section of conclusion and perspective is preferred.

Response 2: We agree with the reviewer concerning the necessity of extending the discussion with a section for conclusions and perspectives. The suggested changes were introduced into the revised version of the manuscript.

Reviewer 2 Report

The manuscript by Rabdano et al., describes the development and characterization of a SARS-CoV-2 vaccine targeting the highly conserved nucleocapsid (N) protein. The authors illustrate the safety and immunogenicity in a variety of species and show that vaccinated hamsters had reduced lung histopathology, lower virus proliferation, lower lung weight relative to the body, and faster body weight recovery compared to controls.

Overall, the paper is well organized and adds to the body of knowledge in the SARS-CoV-2 vaccination field. There are a couple of issues that should be resolved.

Major

Lines 89 and 90. The data do not support the last sentence in the paragraph. The vaccine lessens severity of the disease but done not protect hamsters from disease.

Line 434. Please add the sequence of primers and probe.

Line 611. The date do not support the statement that Convacell protects from infection. The data suggest that the vaccine lessens disease severity.

Minor

Line 229. Please change …collected during euthanasia to …collected after euthanasia.

Materials and methods section.  This section is very detailed which is positive, however some of the animal study text is repeated several times (randomization info, etc.). It would be a good idea to have a separate subsection where the animal work that pertains to all studies is described. It would make the methods section easier to read. I would also place the Animals and Ethics section closer to the start of the methods. Some of the injection and blood volumes seem high for the small mammals so having the ethics statement earlier would help readers know that this volumes were approved prior to the study execution.  

Line 589-592. These sentences should be re-written for clarity.

The paper should be reviewed for grammer. 

Author Response

Response to Reviewer 2 Comments

Point 1: Lines 89 and 90. The data do not support the last sentence in the paragraph. The vaccine lessens severity of the disease but done not protect hamsters from disease.

Response 1: The study of protective effect of vaccination with Convacell® in the Syrian hamster model did indeed show that a mild form of the disease has developed in vaccinated animals and that infection was not completely avoided. The protective effect of nucleocapsid protein is evident from the experimental data and constitutes the main idea of the article. Thus, to make the statement fully supported by the data presented in paper, we have introduced changes into the revised manuscript. The rephrased statements are:

Importantly, we demonstrate that immunization with Convacell® protects Syrian hamsters against development of severe disease after SARS-CoV-2 infection.

At present we can only speculate about the mechanisms how vaccination with Convacell® protects against development of severe disease after SARS-CoV-2 infection because experiments in this direction would be beyond the scope of our study.

Point 2: Line 434. Please add the sequence of primers and probe.

Response 2: The commercial qPCR kit, "Polyvir," produced by Lytech LLC, is certified by the Federal Service for Surveillance in Healthcare, registration No. RZN-2020/9904 dated March 27, 2020. Thus, the quality of kit, including its reagents, specificity, accuracy and other characteristics, is guaranteed via official certification and manufacturer's quality control. The manufacturer did not specify the sequences of the kit's PCR primers and probe in the kit's instruction booklet and refused to provide the information in response to our inquiry. Therefore, unfortunately, we are unable to provide the reviewer with requested primer and probe sequences. To assure that this kit is suitable for our study we would like to state that this kit is the de facto standard kit for SARS-CoV-2 RNA detection in Russia during the ongoing COVID-19 pandemic and can be found in most any clinical laboratory in Moscow, St. Petersburg and other regions of Russia.

Point 3: Line 611. The date do not support the statement that Convacell protects from infection. The data suggest that the vaccine lessens disease severity.

Response 3: see point 1; the changes are inroduced to revised manuscript.

Point 4: Line 229. Please change …collected during euthanasia to …collected after euthanasia.

Response 4: Authors fully agree with the suggested change. It is introduced into the text of the revised manuscript.

Point 5: Materials and methods section.  This section is very detailed which is positive, however some of the animal study text is repeated several times (randomization info, etc.). It would be a good idea to have a separate subsection where the animal work that pertains to all studies is described. It would make the methods section easier to read. I would also place the Animals and Ethics section closer to the start of the methods. Some of the injection and blood volumes seem high for the small mammals so having the ethics statement earlier would help readers know that this volumes were approved prior to the study execution.

Response 5: The improvement suggested by the reviewer is introduced into the text of the revised manuscript: the ethics statement is moved to before the description of experiments on animal models; the repeated descriptions of common animal information and other details are combined together to make text easier to read.

Point 6: Line 589-592. These sentences should be re-written for clarity.

Response 6: Authors agree that the mentioned sentences are not entirely clear. The respective changes to improve clarity are made to the text of revised manuscript.